# Machine Learning Based on Computed Tomography Pulmonary Angiography in Evaluating Pulmonary Artery Pressure in Patients with Pulmonary Hypertension

**DOI:** 10.3390/jcm12041297

**Published:** 2023-02-06

**Authors:** Nan Zhang, Xin Zhao, Jie Li, Liqun Huang, Haotian Li, Haiyu Feng, Marcos A. Garcia, Yunshan Cao, Zhonghua Sun, Senchun Chai

**Affiliations:** 1Department of Radiology, Beijing Anzhen Hospital, Capital Medical University, 2nd Anzhen Road, Chaoyang District, Beijing 100029, China; 2School of Automation, Beijing Institute of Technology, No. 5 Zhongguancun South Street, Haidian District, Beijing 100081, China; 3Department of Pulmonary and Critical Care Medicine, Beijing Anzhen Hospital, Capital Medical University, 2nd Anzhen Road, Chaoyang District, Beijing 100029, China; 4Department of Cardiology, Gansu Provincial Hospital, No. 204, Donggang West Road, Chengguan District, Lanzhou 730099, China; 5Discipline of Medical Radiation Science, Curtin Medical School, Curtin University, Perth 6102, Australia

**Keywords:** artificial intelligence, pulmonary hypertension, computed tomography, artery pressure, diagnosis, automatic assessment, parameter

## Abstract

Background: Right heart catheterization is the gold standard for evaluating hemodynamic parameters of pulmonary circulation, especially pulmonary artery pressure (PAP) for diagnosis of pulmonary hypertension (PH). However, the invasive and costly nature of RHC limits its widespread application in daily practice. Purpose: To develop a fully automatic framework for PAP assessment via machine learning based on computed tomography pulmonary angiography (CTPA). Materials and Methods: A machine learning model was developed to automatically extract morphological features of pulmonary artery and the heart on CTPA cases collected between June 2017 and July 2021 based on a single center experience. Patients with PH received CTPA and RHC examinations within 1 week. The eight substructures of pulmonary artery and heart were automatically segmented through our proposed segmentation framework. Eighty percent of patients were used for the training data set and twenty percent for the independent testing data set. PAP parameters, including mPAP, sPAP, dPAP, and TPR, were defined as ground-truth. A regression model was built to predict PAP parameters and a classification model to separate patients through mPAP and sPAP with cut-off values of 40 mm Hg and 55 mm Hg in PH patients, respectively. The performances of the regression model and the classification model were evaluated by analyzing the intraclass correlation coefficient (ICC) and the area under the receiver operating characteristic curve (AUC). Results: Study participants included 55 patients with PH (men 13; age 47.75 ± 14.87 years). The average dice score for segmentation increased from 87.3% ± 2.9 to 88.2% ± 2.9 through proposed segmentation framework. After features extraction, some of the AI automatic extractions (AAd, RVd, LAd, and RPAd) achieved good consistency with the manual measurements. The differences between them were not statistically significant (t = 1.222, *p* = 0.227; t = −0.347, *p* = 0.730; t = 0.484, *p* = 0.630; t = −0.320, *p* = 0.750, respectively). The Spearman test was used to find key features which are highly correlated with PAP parameters. Correlations between pulmonary artery pressure and CTPA features show a high correlation between mPAP and LAd, LVd, LAa (r = 0.333, *p* = 0.012; r = −0.400, *p* = 0.002; r = −0.208, *p* = 0.123; r = −0.470, *p* = 0.000; respectively). The ICC between the output of the regression model and the ground-truth from RHC of mPAP, sPAP, and dPAP were 0.934, 0.903, and 0.981, respectively. The AUC of the receiver operating characteristic curve of the classification model of mPAP and sPAP were 0.911 and 0.833. Conclusions: The proposed machine learning framework on CTPA enables accurate segmentation of pulmonary artery and heart and automatic assessment of the PAP parameters and has the ability to accurately distinguish different PH patients with mPAP and sPAP. Results of this study may provide additional risk stratification indicators in the future with non-invasive CTPA data.

## 1. Introduction

Pulmonary hypertension (PH) is a malignant pulmonary circulation disease characterized by two aspects: a progressive pulmonary artery pressure increases and a poor natural prognosis. Untreated PH can lead to right ventricular failure due to hypertrophy and remodeling of the right ventricle [1,2]. Further, the median survival in patients with PH without therapy is approximately 2.8 years [2]. Given that PH is a chronic and progressive disease, it should be diagnosed and intervened as early as possible.

PH is defined as a resting mean pulmonary artery pressure (mPAP) of 25 mm Hg or above, and critical PH is defined as 20–25mm Hg [1,3]. Currently, right heart catheterization (RHC) is the only way to measure mPAP accurately. Therefore, a diagnosis of PH is only accepted as confirmed (or excluded) by RHC [4].

A non-invasive and effective method to diagnose PH is essential [5]. Although RHC is the most reliable way to establish the diagnosis of PH, as an invasive procedure, it might be delayed due to its potential complications [6]. In addition, this invasive procedure requires anesthesia so it is not suitable for early screening, and not all the medical institutions have the requirement for RHC inspection. As a non-invasive screening method, which has the advantage of providing more details in high-resolution 3D images [7], computed tomography pulmonary angiography (CTPA) can be routinely used to observe the structure of pulmonary blood vessels, pulmonary parenchyma, and the heart to analyze or exclude possible PH causes. The morphological features in CTPA images serve as important references to assist clinicians in diagnosing PH. Assuming that the morphological features hidden in CTPA images are further excavated, we might be able to build a regression model to predict the PAP value based on CTPA images.

Using CTPA images to assist physicians in risk stratification of patients in PH is another meaningful study. Previous studies have shown that chronic obstructive pulmonary disease (COPD) is one of the most common causes of PH [8,9]. Although patients with COPD often present mild or moderate PH, typically with mPAP between 20 and 40 mm Hg, mild mPAP elevations can also lead to poor prognosis [9]. Patients with severe PH are defined as having a resting mPAP > 35 to 40 mm Hg, and a sharp increase in mPAP often leads to right ventricular failure. Moreover, these patients often have severe hypoxemia and exhibit hypocapnia, reducing their life expectancy [10]. In addition, Nadrous et al. [11] proved that idiopathic pulmonary fibrosis patients with pulmonary artery systolic pressure (sPAP) of >50 mm Hg had a higher mortality with evidence of 1-year and 3-year mortalities of 56% and 68%. Further, in patients with ≥moderate primary mitral regurgitation [12], the risk of adverse events is higher for patients of SPAP ≥ 55 mmHg. Another study pointed out that in patients with asymptomatic primary mitral valve regurgitation or flail leaflet, the prognostic value of peak exercise sPAP ≥ 50 mm Hg is significant [13]. Therefore, it is of great significance to explore the possibility to classify patients in PH based on mPAP of 40 mm Hg and sPAP of 55 mm Hg by non-invasive measurements in order to carry out risk stratification of PH patients.

Artificial intelligence (AI) has great potential in cardiovascular function assessment. One of the major applications of AI in this field is to assess functional parameters from morphological features in CTPA of patients, which includes constructing regression models and classification models to help with the diagnosis of PH. Liu et al. [14,15] has demonstrated that some specific cardiovascular parameters of CTPA could be the predictors to assess chronic thromboembolic pulmonary hypertension (CTEPH) severity and right ventricular function. Melzig et al. [16] pointed out that some morphological parameters of CTPA such as 3D volumes are valuable in noninvasively estimating pulmonary artery functional parameters, which means that CTPA data are beneficial for noninvasively predicting PH. Dong et al. [17] claimed that several morphological parameters of CTPA performed well in evaluating the severity of right ventricular dysfunction. Beyond the pulmonary field, in the study that shows the performance of coronary CT angiography-derived fractional flow reserve (CT-FFR) in diagnosing ischemia in myocardial bridging (MB), Zhou et al. [18] found that machine learning-based coronary CT-FFR showed good diagnostic performance compared to invasive FFR.

In this study, aiming to evaluate the value of CTPA morphological data analysis in the diagnosis of PH, we proposed a methodology to extract and select the morphological features from CTPA images, and then build regression and classification models based on machine learning methods. Using this methodology, we implemented a workflow for the auxiliary diagnosis of PH based on the analysis of CTPA images.

## 2. Materials and Methods

The overall workflow of the proposed method is shown in Figure 1. After the acquisition and selection of the study population, CTPA data and RHC data of the corresponding patients were both collected and recorded. Then, four main steps were followed to finish the work. First, we accomplished the annotation task via computer software and the segmentation of eight selected substructures of the heart and pulmonary from the CTPA images using a proposed segmentation framework. After that, we extracted the morphological features and their second-order features within the substructures for subsequent analysis. To establish a better regression model to predict the mPAP, sPAP, pulmonary artery diastolic pressure (dPAP), and total pulmonary resistance (TPR) of a specific patient, we selected some morphological features which are highly correlated with the pulmonary arterial pressure and reduced the dimensions of the selected features matrix. Then, we built a regression model to predict the PAP parameters and implemented a series of statistical analysis to verify the consistency between predicted and true values. In addition, we used 10-fold cross-validation to establish a classification model to determine whether a patient has severe pulmonary arterial pressure or not based on mPAP and sPAP.

### 2.1. Patients

A total of 55 patients with PH were retrospectively enrolled in this study between June 2017 and July 2021 from a single tertiary center of Beijing Anzhen Hospital. PH was diagnosed as a mean pulmonary artery pressure ≥25 mm Hg at rest as evaluated by right-heart catheterization (RHC) [19]. The PH types were defined as the 5-classification standard [20]. Patients who received CTPA and right-heart catheterization exam within 1 week were included. Demographic and clinical data were retrieved from each patient’s electronic medical record. Exclusion criteria included the following: (i) the interval of CTPA and RHC exam longer than 1 week; (ii) inferior image quality of CTPA; and (iii) inadequate clinical data acquired from the electronic medical record system. The study was approved by the local institutional ethics review committee. Informed consent was waived because of the retrospective study.

### 2.2. Imaging Protocol

A 320-row CT system (Aquilion ONE, Toshiba, Otawara, Japan) was used in all patients for CTPA scanning. All the exams were performed with non-ECG-gated helical scan protocol. Patients were positioned supine and feet first into the gantry. Dual scanograms were used for determination of the anatomical coverage. The volume was placed to cover the entire lung fields from the pulmonary apex to the posterior costophrenic angle. Each volume CTPA data acquisition was acquired with a single breath-hold. The CT gantry rotation time was 330 ms. The tube voltage was 100–120 kV; effective tube current was 200–300 mA adjusted by personal body mass index (BMI). The collimation was 0.625 mm; pitch was 0.99. All the data were reconstructed using a standard soft-tissue and lung kernel (FC56). Images were reconstructed with slice thickness of 0.9 mm, interval of 0.45 mm.

A total of 40–50 mL contrast medium (Omnipaque 350, GE Healthcare, Shanghai, China) was intravenously injected by using a dual-head power injector with the injection rate of 3.5–4.5 mL/s adjusted according to BMI and the CT data acquisition time. A saline chaser bolus of 30 mL was injected with the same rate as the contrast medium. A region of interest was placed at the level of the main pulmonary artery for bolus tracking. The exposure was triggered with a 5 s delay after the 150 HU threshold was reached.

### 2.3. Right-Heart Catheterization

Right-heart catheterization was performed with the Seldinger technique. Under X-ray fluoroscopic guidance, an 8F Swan-Ganz catheter (Baxter Healthcare, Irvine, CA, USA) was induced through the right internal jugular vein. After 10 min rest, the hemodynamic parameters including sPAP, dPAP, pulmonary capillary wedge pressure (PCWP), and cardiac output (CO) were obtained at end-expiration. The mPAP and pulmonary vascular resistance (PVR) were calculated.

### 2.4. Segmentation Ground-Truth Definition

#### 2.4.1. Data Standardization and the Localization of Heart and Pulmonary Artery

CTPA images were all automatically cropped into 512 × 512 × 480 pixels, which included the full volume from the top of the main pulmonary artery to the bottom of the right ventricle.

#### 2.4.2. Contours of Eight Substructure Delineation

The preprocessed images were segmented into eight substructures with the use of computer-assisted tools by an experienced radiologist (N.Z., with 9 years of experience in CTPA image interpretation). These compartments included: (1) left ventricle blood cavity (LV); (2) right ventricle blood cavity (RV); (3) left atrium blood cavity (LA); (4) right atrium blood cavity (RA); (5) left pulmonary artery (LPA); (6) right pulmonary artery (RPA); (7) main pulmonary artery (MPA); and (8) the ascending aorta (AA). Following manual delineation, annotations were visually inspected and manually retraced as needed by another experienced radiologist for quality checking (J.Y.L., with 25 years of experience in cardiovascular imaging) [21].

### 2.5. Image Segmentation

Prior to training our model, the dataset was randomly split into training (80%; 44 of 55) and test (20%; 11 of 55) sets. After data preprocessing, the training set was further divided into training (80%; 35 of 44) and validation (20%; 9 of 44) sets. We adopted a deep learning framework referred to as a nnU-Net [22] to perform segmentation on the preprocessed data. Then, we selected a full resolution 3D U-Net architecture to perform segmentation. Using this architecture, eight substructures of the lungs and heart were segmented, with four substructures for each part. Therefore, two independent 3D U-Net networks were trained for segmentation. The convolutional neural network architecture is shown in Figure 2. The batch-size is 2 and the patch-size is 128 × 128 × 128. In addition, the loss function is a combination of the cross-entropy loss function and the Dice loss function.

However, some of the outputs were larger or smaller than the ground-truth due to the low contrast between the segmentation part and the adjacent structures, which is mainly caused by the lack of contrast agent. To optimize these segmentation results, we proposed a framework to create a better mask for poor results. First, we selected the axiomatic good results as positive masks and the others as negative masks. Second, we expanded and contracted every mask slightly to obtain a group of masks including the current, the expanded, and the contracted one. Then, the image intensities of the mask contours were extracted and flattened into a one-dimension vector. After zero padding, the vectors were concatenated and then fitted into a self-attention module followed by a fully connected neural network which determined whether the input was a positive mask or not. The program would continuously resize the input mask until the output became positive mask. Finally, all the segmentations were used to automatically and reliably measure the morphological features, thus putting them into predictive models.

### 2.6. Features Extraction and Selection

#### 2.6.1. The Extraction of Morphological and Second-Order Features

Since it is proven that pulmonary artery and ventricles are strongly correlated with pulmonary arterial hypertension, the morphological features of all the eight substructures and their second-order features may contain direct correlations to the medical metrics such as the mean pulmonary artery pressure. Therefore, we obtained as many morphological features and their second-order features from the segmentations as possible. Liu et al. [14] demonstrated that some specific morphological parameters of CTPA could be used to assess chronic thromboembolic pulmonary hypertension (CTEPH).

#### 2.6.2. Feature Selection and Dimension Reduction

As shown in Figure 3, the following four methods were used to reduce data redundancy and select meaningful morphological features and second order features. First, the Spearman test was used to pick up significant features. Second, the Pearson correlation coefficient r between any two features was calculated. When r > 0.9, the feature with a lower Pearson correlation coefficient with the dependent variable was removed. Third, we normalized the features to a range of data with mean 0 and variance 1. This can be implemented with StandardScaler in Python scikit-learn environment. Finally, the principal components analysis was adopted to reduce the dimension of the features in an unsupervised way. Therefore, we selected reasonable morphological features and second-order features.

### 2.7. Regression and Classification Model Construction

We used statistical analysis to compare the manual and automated measurement methods and to verify the correlation of the two measurements. At the same time, we also compared the performance of different models on AI automatic measurement features, and finally selected the XGBoost model to complete regression and classification. After comparing with the SVM model and CatBoost model, the features of 55 patients were used as input for the XGBoost [23] regression and classification model. The PAP parameters the regression model aimed to predict included mPAP, sPAP, dPAP, and TPR, while the classification model only used mPAP and sPAP to stratify the risk of patients with PH. The max depth of the model is set as 3. We completed 20 times ten-fold cross validation to ensure the validity of the results stratified on patient level [24]. Finally, we proved that our workflow could assess functional parameters through CTPA to help the auxiliary diagnosis of PAH.

### 2.8. Statistics Analysis

All statistical analyses were performed using SPSS ver. 27.0 (SPSS Inc., Chicago, IL, USA). We used dice score to evaluate the performance of the proposed segmentation framework compared with the nnU-Net. After that, we compared the AI automatic measurements and manual measurements through Paired *t*-test and Bland–Altman analyses. Then, the correlation of morphological features with mPAP, sPAP, dPAP, and TPR was determined by the Spearman test. After that, the statistical differences of selected features were determined by the Pearson test. Using RHC results as the ground truth, intraclass correlation coefficients (ICC) between mPAP, sPAP, dPAP, and TPR measured by RHC and the output of our regression model were calculated. After that, we set the classification threshold as 40mm Hg and 55 mm Hg for mPAP and sPAP to separate the patients into different risk classes, respectively. Then, we calculated sensitivity and specificity of the patient risk level derived from RHC results with the output of our classification model. In addition, the receiver operating characteristic curve (ROC) was used to evaluate the classification performance.

## 3. Results

### 3.1. Study Population Characteristics

A total of 55 patients with PH were included in this study with clinical characteristics shown in Table 1. Chronic pulmonary embolism (15/55, 27.27%) was the most common reason of pulmonary hypertension in this study. There was only 1 patient diagnosed with mPAP ≥ 70 mm Hg.

### 3.2. Computational Time

The average training time of the segmentation network on each patient is about 6 min. During the testing, the average inference time of the network model on each patient is about 10 min. The training and testing of the segmentation network were all done on NVIDIA A100 Tensor Core GPU. Moreover, the average time for feature extraction on each case is about 17 s. In the prediction phase, the computational time is about 8 s for the regression and classification for each case. The feature extraction and prediction of the regression and classification model were all tested on AMD Ryzen 7 5800H with Radeon Graphics CPU.

### 3.3. The Performance of the Segmentation Framework

Using an independent testing dataset, the average dice score for segmentation is 87.3% ± 2.9 with original nnU-Net. However, from Table 2, we can see that the dice score of each part improved significantly and the average dice score becomes 88.2% ± 2.9. Apart from dice score, the improvements can be seen in Figure 4. Compared with the result of proposed network framework in Figure 4c, myocardium of LV is mis-segmented as chamber of LV using nnU-Net in Figure 4b which overestimates the area of LV. In addition, the MPA segmentation mask which does not cover the original pulmonary artery in Figure 4e is slightly expanded to a plausible mask in Figure 4f.

### 3.4. Comparison of Manual Measurement and AI Automatic Measurement

Table 3 shows that the differences in RPAd, AAd, RVd, and LAd measured by the manual measurement and the AI automatic measurement were not statistically significant (t = 1.222, *p* = 0.227; t = −0.347, *p* = 0.730; t = 0.484, *p* = 0.630; t = −0.320, *p* = 0.750, respectively). However, the results also suggest that the differences in MPAd, LPAd, LVd, and RAd measured by the manual the AI automatic measurements were statistically significant (t = −3.573, *p* = 0.001; t = 4.394, *p* < 0.001; t = 4.255, *p* < 0.001; t = −7.096, *p* < 0.001; respectively). Nevertheless, according to the correlation between these features and the PAP parameters, only LVd is valuable for regression and classification. Then, Bland–Altman analyses [25] for features assessed by manual and AI automatic measurements were carried out, and the corresponding biases (limits of agreement) of MPAd, RPAd, LPAd, AAd, LVd, RVd, LAd, and RAd are −2.25 mm (−11.4 mm, 6.90 mm), 0.575 mm (−6.28 mm, 7.43 mm), 2.84 mm (−6.56 mm, 12.25 mm), −0.15 mm (−6.45 mm, 6.15 mm), 5 mm (−12.18 mm, 22.18 mm), 0.52 mm (−15.27 mm, 16.31 mm), −0.27 mm (−12.86 mm, 12.31 mm), and −11.68 mm (−35.83 mm, 12.46 mm), respectively. Figure 5 shows Bland–Altman analyses between manual and AI measurements.

### 3.5. The Correlations between Pulmonary Artery Pressure and Selected Features

The correlations between morphological features and their second-order features obtained by automatic measurement and pulmonary artery pressure obtained by RHC are shown in Table 4. The table lists some of the characteristics and four pulmonary artery pressure values that are highly correlated with each other: mean pulmonary artery pressure, pulmonary artery systolic pressure, pulmonary artery diastolic pressure, and total pulmonary resistance. Correlations between pulmonary artery pressure and CTPA features show a positive correlation between mPAP and RAd/LAd (r = 0.333, *p* = 0.012), and a negative correlation between mPAP and LAd, LVd, LAa (r = −0.400, *p* = 0.002; r = −0.208, *p* = 0.123; r = −0.470, *p* = 0.000; respectively), but no correlation was found between mPAP and MPAd, MPAd/AAd, RPAd, LPAd, RVd, RAd.

### 3.6. The Performance of the Regression Model

In the regression task, we compared three commonly used regression models including XGBoost, CatBoost, and SVM. As shown in Figure 6, with the same testing dataset of patients in mPAP regression task, XGBoost (MSE = 12.81) performed better than SVM (MSE = 60.24) and CatBoost (MSE = 28.37). Thus, we chose XGBoost regressor to predict the sPAP, dPAP, and TPR values of patients in this study. Figure 6 shows that the MSE of the results on dPAP, sPAP, and TPR is 16.94, 21.88, and 69,862.59, respectively. To demonstrate the consistency between the predicted and true values with different models and pressure types, we calculated the ICC of these two groups. As shown in Table 5, assuming the interaction effect is absent, the ICC of CATBoost regressor, SVM regressor, and XGBoost regressor to predict mPAP value is 0.689, 0.138, and 0.934, respectively. Moreover, the ICC of XGBoost regressor to predict sPAP, dPAP, and TPR is 0.981, 0.903, and 0.685, respectively. The above results show that the XGBoost regressor is able to predict mPAP, sPAP, and dPAP on a small dataset of adult patients with PH.

### 3.7. The Performance of the Classification Model

In the classification task, we also compared three commonly used classification models, namely XGBoost, CatBoost, and SVM. Setting the cut-off value as 40 mm Hg, we tested the above three different machine learning models to classify patients with PH based on mPAP. As shown in Figure 7, using the same testing dataset of patients in mPAP classification task, XGBoost (AUC = 0.911, *p* < 0.001) performed better than SVM (AUC = 0.679, *p* = 0.2846) and CatBoost (AUC = 0.893, *p* < 0.001). Similarly, we used these three models to classify patients with PH by sPAP with a cut-off value of 55 mm Hg. The results are also shown in Figure 6, yielding AUC of 0.556 (*p* = 0.8257), 0.639 (*p* = 0.6262) and 0.833 (*p* = 0.0057) for CatBoost, SVM and XGBoost, respectively.

## 4. Discussion

In this study, we developed a fully automated CTPA image-based framework for the additional diagnosis of PH. First, this framework can achieve the segmentation of eight substructures of pulmonary artery and heart (LV, RV, LA, RA, LPA, RPA, MPA, and AA). Using an independent testing dataset, the average Dice score for segmentation with our proposed framework could reach 88.2%. Second, we completed the features extraction based on the segmentation outcome. Some of the AI automatic extractions (AAd, RVd, LAd, and RPAd) achieved good consistency with the manual measurements. However, the differences between MPAd, LPAd, LVd, and RAd, respectively measured by AI and physicians, are statistically significant (*p* < 0.001). Then, we selected morphological features or their second-order features with high correlations between mPAP, sPAP, dPAP, and TPR, and achieved features dimension reduction using principal component analysis. Finally, the regression model for predicting mPAP, sPAP, dPAP, and TPR and the classification model for separating patients with PH by mPAP or sPAP with different risk levels were executed. Good consistency existed between the outcome of the regression model predicted mPAP, dPAP, and sPAP and the ground-truth from RHC (ICC = 0.934, *p* = 0.002; ICC = 0.903, *p* = 0.006; ICC = 0.981, *p* = 0.000, respectively). The AUC of ROC curve of the classification model reached 0.911 (*p* < 0.001) for mPAP data and 0.833 (*p* = 0.0057) for sPAP data.

The segmentation for the eight substructures were based on our proposed framework. The part after pre-segmentation of this framework is a complement which relies on the performance of nnU-Net based pre-segmentation. The proposed method utilizes the image intensity information near the edge of the mask to obtain the feature map through an attention mechanism. The feature map that aims to represent the drastic changes between the segmentation part and the adjacent area is a useful input for the classification neural network. However, the pre-segmentation outputs for some parts may appear as unpredictable results, such as a sudden disappearance in a certain plane or the results appear as two separate pieces. These problems cannot be solved by current framework, so it is necessary to propose a better segmentation method combining latest knowledge in machine learning field.

In this study, the morphological features contained in CTPA images were extracted based on 3D image segmentations. Previous studies have shown that some morphological parameters manually measured by physicians have high correlations between pulmonary vascular resistance in patients with chronic thromboembolic pulmonary hypertension [7,14,15]. However, Liu’s study did not achieve automatic extraction of morphological parameters, thus lack of repeatability. Our study is more efficient and ensures strong repeatability through automatic segmentation and morphological parameter extraction. Nevertheless, the present network based on nnU-Net for segmentation cost about 10 min to inference. Therefore, we may try to improve the network’s segmentation performance and prediction speed by improving the network structure or using methods such as model pruning.

The analysis of morphological features in this study represents mainly an extension of the existing research. Liu [14] and Jia [17] et al. proposed that not only the morphological features themselves but also the ratio between them were correlated to the mPAP levels or right ventricular dysfunction. Melzig et al. [16] used the volume of main, right, and left pulmonary arteries and combined echocardiographic sPAP to achieve a higher accuracy for the prediction of mPAP. Our study included one-dimensional and two-dimensional morphological features of heart and pulmonary arteries and the ratio between selected features, such as RAd/LAd, LVd, and LAa. In recent years, radiomics has gradually emerged, which can extract many invisible shapes, textures, and image intensity features. Cetin et al. [26] demonstrated the feasibility and the clinical value of the cardiac MRI radiomics in analyzing the cardiovascular risk factors including diabetes, hypertension, high cholesterol, and smoking. Moreover, Lu et al. [27] proposed a combined model, including morphological features and radiomics features from CT scan, to distinguish minimally invasive adenocarcinomas and invasive adenocarcinomas, with high potential to provide for the auxiliary diagnosis. Therefore, if hundreds of thousands of features could be extracted from images by radiomics, and features extraction and dimension reduction can be performed on them, then the performance of the regression and classification model in our study may be better.

The regression model proposed in this study is designed to assess pulmonary arterial pressure in the diagnosis of PH based on CTPA images. The ICC coefficient between the predicted value and the actual value of this regression model of mPAP, sPAP, and dPAP is 0.934, 0.903, and 0.981, respectively. The regression model results demonstrate the potential feasibility of inferring the functional parameters of patients based on morphological features obtained from CTA images without RHC. However, the number of samples in this study is small, and all of them are adult patients with PH. There is a lack of samples from different groups, such as people without PH and children in PH. Therefore, it is necessary to conduct research in the future with inclusion of larger and more diverse samples to verify the generalizability of the results of this model.

The classification model used in this study uses 40 mm Hg and 55 mm Hg as the cut-off value of mPAP and sPAP to characterize PH patients into two categories. The classification model is a further extension and application of the regression model proposed in this study, and has a certain potential to achieve risk stratification for patients with PH. However, since the samples in this study did not contain patients without PH, the ability of the model to discriminate between PH and non-PH patients remains to be determined. Therefore, patients without PH can be introduced to construct a classification model capable of diagnosing PH in the future. On the other hand, the mPAP values of patients with PH in this study are concentrated and the lower sPAP values are not enough, resulting in unbalance of the samples. In the future, we may construct a large number of samples of patients with a balanced distribution of mPAP and sPAP values. Then we may be able to complete the multi-classification of patients in PH.

At present, the classification model proposed in this study focuses on the auxiliary diagnosis. For a better realization of a pathological clear division of patients with pulmonary arterial hypertension (pulmonary hypertension caused by left heart disease, pulmonary disease and hypoxia, chronic thromboembolic pulmonary hypertension, and unknown multi-factor mechanism), it is necessary to further improve the diversity of collected samples and integrate more information contained in multimodal medical images into the analysis, therefore achieving the purpose of pathological classification for PH. As discussed previously, our study was limited in analyzing one-dimensional and two-dimensional morphological features of heart and pulmonary arteries and the ratio between the selected features. In future study, we will explore the volumetric information of heart and pulmonary artery morphology as well as the spatial relationship between different intra- and extra-cardiac structures to improve the accuracy of PAP parameter evaluation.

## 5. Conclusions

In this study, we developed a fully automatic framework for supplementary diagnosis of PH based on CTPA images. We achieved automatic segmentation of eight substructures (LV, RV, LA, RA, LPA, RPA, MPA, and AA) of the pulmonary artery and the heart with high accuracy by our proposed segmentation framework. Based on this segmentation result, the extraction of morphological features was automatically carried out by the machine learning, which was highly repeatable. Our results showed that it is feasible to assess PAP parameters in patients with PH from CTPA images rather than invasive RHC examinations. Furthermore, our proposed framework can also perform a preliminary classification of PH patients, which is a contribution to the diagnosis or management of PH patients.

## Figures and Tables

**Figure 1 jcm-12-01297-f001:**
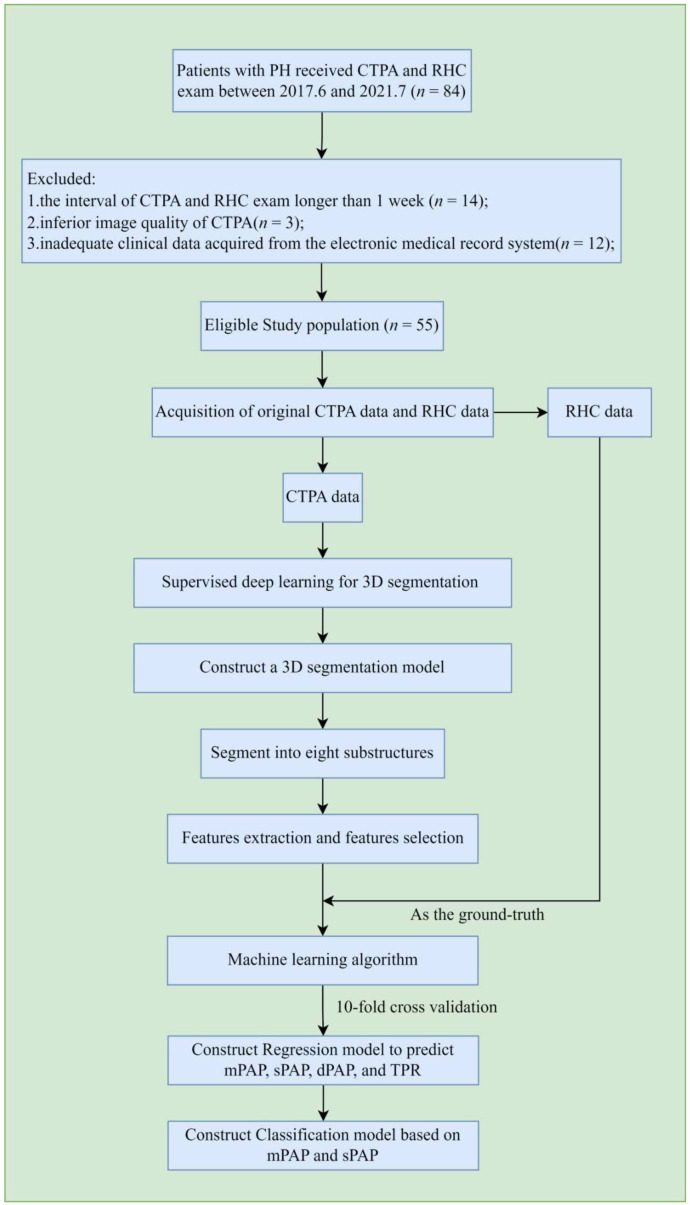
The overall workflow. After the selection and exclusion of patient population and CTPA images acquisition, eight substructures of heart and pulmonary arteries were automatically segmented as regions of interest (ROIs). Morphological features and their second-order features are extracted from these ROIs. Then, the essential features were selected through statistical methods and then reconstructed to reduce the redundancy of the features matrix. In the last step, meaningful features and corresponding results are set as an input through machine learning algorithms to obtain a regression model and a classification model.

**Figure 2 jcm-12-01297-f002:**
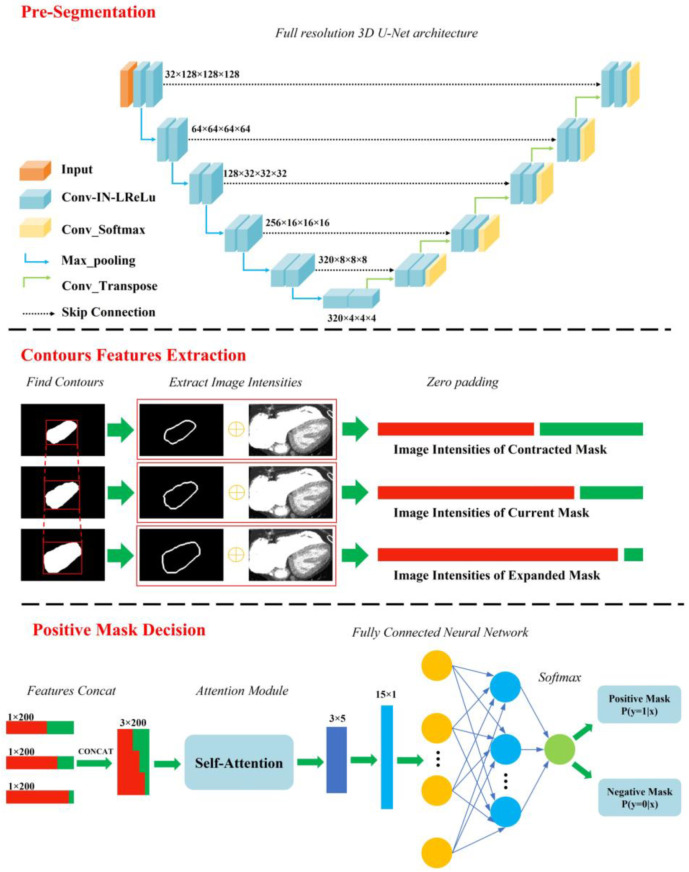
Segmentation framework architecture for accurate segmentation of the substructures of pulmonary artery and heart, respectively. The proposed segmentation framework consists of full-resolution 3D U-Net; contours feature extraction; and positive mask decision.

**Figure 3 jcm-12-01297-f003:**
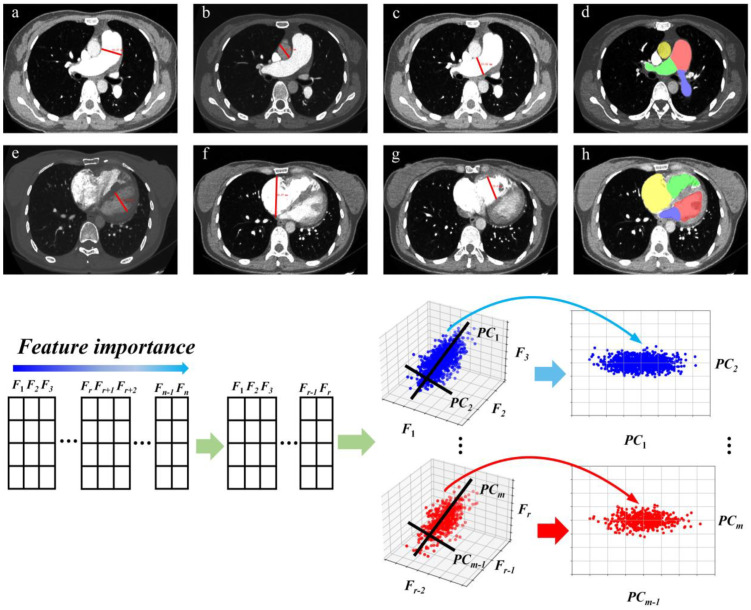
Some of the morphological features on CTPA are shown above. (**a**) Diameter of main pulmonary artery (MPAd), (**b**) diameter of ascending aorta (AAd), (**c**) diameter of right pulmonary artery (RPAd), (**d**) area of LPA (blue), RPA (green), MPA (red), and AA (yellow), (**e**) diameter of left ventricle, (**f**) diameter of right atrium, (**g**) diameter of right ventricle, (**h**) area of LV (red), RV (green), LA (blue), and RA (yellow). They were all assessed on transversal images in the diastole. The features extracted from the original CTPA images were first organized in a table, sorted by the Spearman’s rank correlation coefficient between one feature and mean pulmonary artery pressure. Since any two features we selected may contain a strong correlation, we removed redundant features by calculating the Pearson coefficient of all features with each other. Then we selected r features from all features we obtained. The considerately selected features were then normalized and feature dimension reduction was performed by principal component analysis. Finally, we chose m features for the regression and classification model.

**Figure 4 jcm-12-01297-f004:**
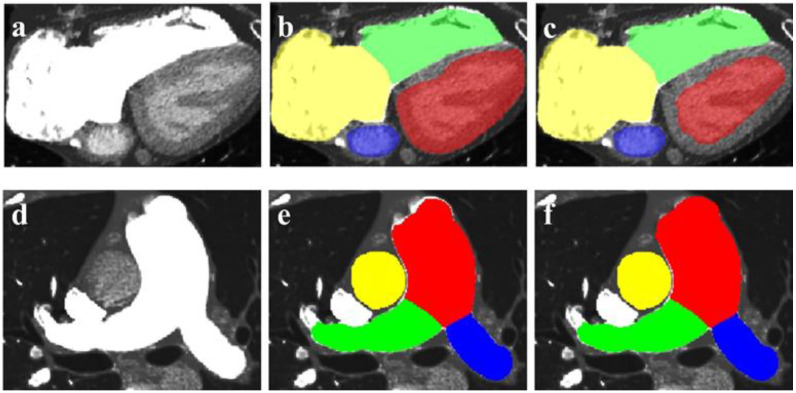
The performance of the proposed network framework. (**a**,**d**) are the original image of heart and pulmonary artery, respectively; (**b**,**e**) are the segmentation outputs of nnU-Net; (**c**,**f**) are the segmentation outputs of the proposed network framework.

**Figure 5 jcm-12-01297-f005:**
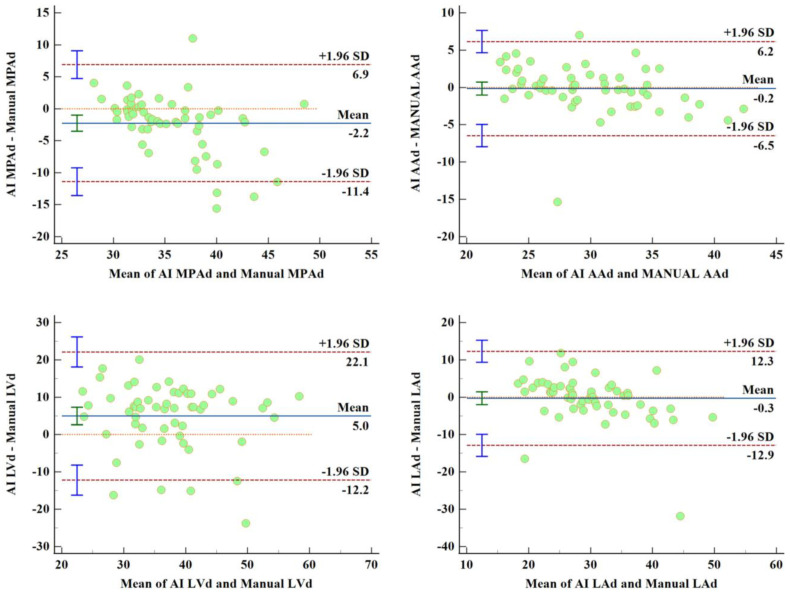
Bland–Altman analyses for features assessed by AI automatic and manual measurements show that the metrics measured by our automatic measurement method are in accordance with the ground-truth measured manually by experienced physicians.

**Figure 6 jcm-12-01297-f006:**
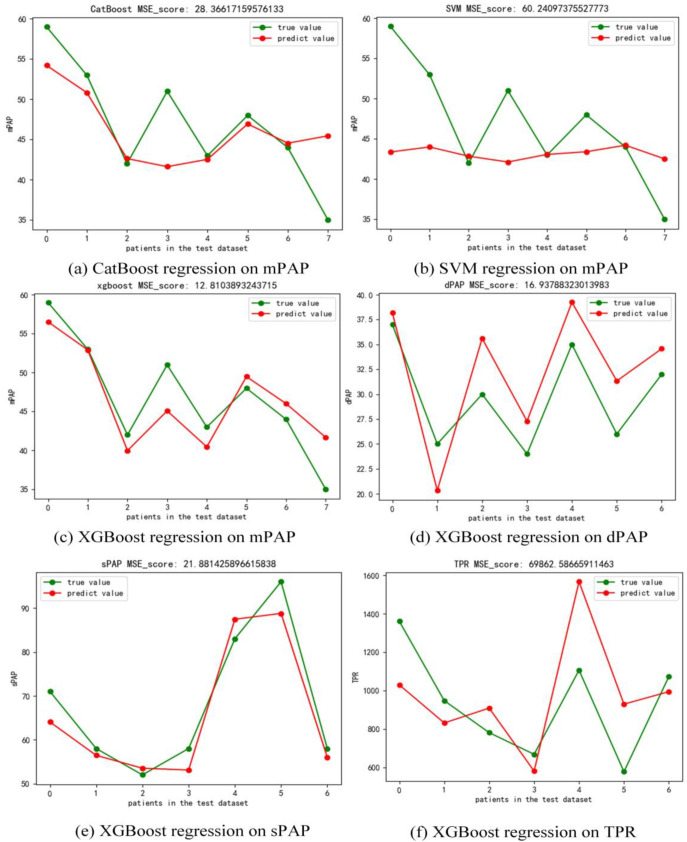
(**a**) CatBoost regression on mPAP (MSE = 28.37), (**b**) SVM regression on mPAP (MSE = 60.24), (**c**) XGBoost regression on mPAP (MSE = 12.82), (**d**) XGBoost regression on dPAP (MSE = 16.94), (**e**) XGBoost regression on sPAP (21.88), (**f**) XGBoost regression on TPR (MSE = 69,862.59).

**Figure 7 jcm-12-01297-f007:**
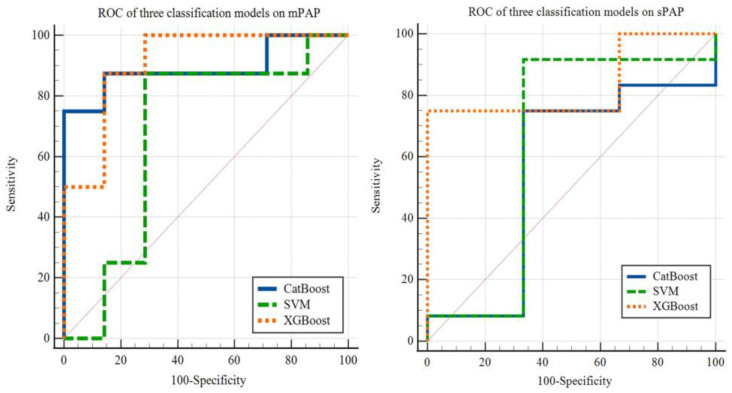
Area under the receiver operating characteristic curve (AUC) shows XGBoost model performs better than CatBoost and SVM model. The AUC of XGBoost model classification on mPAP and sPAP is 0.911 (*p* < 0.001) and 0.833 (*p* = 0.0057), respectively.

**Table 1 jcm-12-01297-t001:** Clinical and hemodynamic characteristics of patients with PH.

Characteristics		Values
Baseline Parameters (Mean ± SD)		
	Age (years)	47.75 ± 14.87
	Disease duration (years)	3.55 ± 7.02
	Gender (male/female)	13/42
	Body mass index (kg/m^2^)	23.94 ± 3.87
	Body surface area (m^2^)	1.65 ± 0.15
	BNP (pg/mL)	289.67 ± 268.97
Clinical classification of PH (*n*)		
	I	12 (21.8%)
	II	6 (10.9%)
	III	10 (18.2%)
	IV	15 (27.3%)
	V	12 (21.8%)
Class of mPAP (*n*)		
	Mild (25–39 mm Hg)	21 (38.2%)
	Severe (≥70 mm Hg)	34 (61.8%)
Class of sPAP (*n*)		
	Mild (44–55 mm Hg)	8 (14.5%)
	Severe (≥55 mm Hg)	47 (85.4%)
NYHA classification (*n*)		
	I	11 (20.0%)
	II	20 (36.4%)
	III	22 (40.0%)
	IV	2 (3.6%)
Hemodynamics (mean ± SD)		
	sPAP (mmHg)	68.35 ± 20.62
	dPAP (mmHg)	35.65 ± 42.53
	mPAP (mmHg)	42.82 ± 12.74
	CO (l/min)	4.64 ± 2.54
	PCWP (mmHg)	6.86 ± 4.79
	PVR (dynes.sec.mm^−5^)	979.03 ± 594.73

Note. BNP, B-type natriuretic peptide; CO, cardiac output; dPAP, pulmonary artery diastolic pressure; mPAP, mean pulmonary artery pressure; NYHA, New York Heart Association; PCWP, pulmonary capillary wedge pressure; PH, pulmonary hypertension; PVR, pulmonary vascular resistance; sPAP, pulmonary artery systolic pressure.

**Table 2 jcm-12-01297-t002:** Dice score between the original nnU-Net and the proposed framework.

	LV	RV	LA	RA	MPA	LPA	RPA	AA
nnU-Net	82.4 ± 7.8	86.9 ± 6.8	86.9 ± 5.2	88.9 ± 10.3	91.6 ± 5.5	90.8 ± 2.9	86.4 ± 5.1	84.5 ± 9.0
Proposed framework	83.2 ± 7.1	87.0 ± 7.0	87.4 ± 4.7	90.7 ± 5.8	92.5 ± 3.4	91.3 ± 2.4	86.9 ± 4.6	86.4 ± 8.0

Note. The values show the mean (±standard deviation) of all images from the CTPA for each segmentation label. AA, ascending aorta; LA, left atrium; LPA, left pulmonary artery; LV, left ventricle; MPA, main pulmonary artery; RPA, right pulmonary artery; RA, right atrium; RV, right ventricle.

**Table 3 jcm-12-01297-t003:** Comparison of manual measurement and AI automatic measurement (paired samples *t*-test).

Features	The Difference of Manual and AI Automatic Measurement				AI Automatic Measurement	Manual Measurement
	Mean	Test Statistic *t*	95% CI of Difference	Sig.	Mean ± SD	Mean ± SD
MPAd (mm)	−2.249	−3.573	−3.511 to −0.9874	0.001	34.48 ± 3.98	36.73 ± 5.98
RPAd (mm)	−0.5759	1.222	−0.3693 to 1.521	0.227	24.33 ± 3.01	23.75 ± 3.82
LPAd (mm)	2.843	4.394	1.546 to 4.140	0.000	25.42 ± 4.61	22.57 ± 3.73
AAd (mm)	−0.1506	−0.347	−1.020 to 0.718	0.730	29.81 ± 4.61	29.96 ± 5.57
LVd (mm)	4.975	4.255	2.632 to 7.319	0.000	39.78 ± 8.51	34.80 ± 9.66
RVd (mm)	0.5214	0.484	−1.636 to 2.679	0.630	44.39 ± 5.62	43.87 ± 9.32
LAd (mm)	−0.2745	−0.320	−1.994 to 1.445	0.750	29.60 ± 6.52	29.88 ± 9.06
RAd (mm)	−11.68	−7.096	−14.98 to −8.382	0.000	46.47 ± 7.49	58.15 ± 15.15

Note. AAd, diameter of ascending aorta; LAd, diameter of left atrium; LPAd, diameter of left pulmonary artery; LVd, diameter of left ventricle; MPAd, diameter of main pulmonary artery; RAd, diameter of right atrium; RPAd, diameter of right pulmonary artery; RVd, diameter of right ventricle.

**Table 4 jcm-12-01297-t004:** Correlation between pulmonary artery pressure and CTPA features by Spearman test.

	Mean ± SD (*n* = 55)	mPAP (*n* = 55)	sPAP (*n* = 54)	dPAP (*n* = 54)	TPR (*n* = 48)
MPAd (mm)	34.48 ± 3.98	r = −0.107 (*p* = 0.435)	r = −0.185 (*p* = 0.180)	r = 0.011 (*p* = 0.938)	r = −0.115 (*p* = 0.435)
RPAd (mm)	24.33 ± 3.01	r = −0.149 (*p* = 0.277)	r = −0.115 (*p* = 0.410)	r = −0.065 (*p* = 0.640)	r = −0.250 (*p* = 0.087)
LPAd (mm)	25.42 ± 4.61	r = −0.120 (*p* = 0.381)	r = 0.050 (*p* = 0.721)	r = −0.156 (*p* = 0.261)	r = −0.170 (*p* = 0.249)
AAd (mm)	29.81 ± 4.61	r = −0.265 (*p* = 0.050)	r = −0.279 (*p* = 0.041) *	r = −0.183 (*p* = 0.186)	r = −0.156 (*p* = 0.289)
MPAd/AAd	1.17 ± 0.17	r = −0.212 (*p* = 0.121)	r = 0.175 (*p* = 0.216)	r = 0.208 (*p* = 0.131)	r = 0.091 (*p* = 0.539)
RPAd/LPAd	0.988 ± 0.235	r = −0.034 (*p* = 0.806)	r = −0.194 (*p* = 0.161)	r = 0.051 (*p* = 0.731)	r = 0.004 (*p* = 0.978)
LVd (mm)	39.78 ± 8.51	r = −0.208 (*p* = 0.123)	r = −0.348 (*p* = 0.009) **	r = −0.183 (*p* = 0.181)	r = −0.107 (*p* = 0.470)
RVd (mm)	44.39 ± 5.62	r = 0.227 (*p* = 0.092)	r = 0.243 (*p* = 0.074)	r = 0.207 (*p* = 0.129)	r = 0.195 (*p* = 0.185)
RVd/LVd	1.17 ± 0.34	r = 0.302 (*p* = 0.023) *	r = 0.416 (*p* = 0.002) **	r = 0.270 (*p* = 0.046) *	r = 0.259 (*p* = 0.075)
LAd (mm)	29.60 ± 6.52	r = −0.400 (*p* = 0.002) **	r = −0.351 (*p* = 0.009) **	r = −0.318 (*p* = 0.018) **	r = −0.453 (*p* = 0.001) **
RAd (mm)	46.47 ± 7.49	r = 0.024 (*p* = 0.859)	r = −0.058 (*p* = 0.675)	r = 0.038 (*p* = 0.783)	r = 0.182 (*p* = 0.216)
RAd/LAd	1.64 ± 0.45	r = 0.333 (*p* = 0.012) *	r = 0.203 (*p* = 0.138)	r = 0.328 (*p* = 0.015) *	r = 0.512 (*p* = 0.000) **
LAa (mm^2^)	1746.44 ± 621.63	r = −0.470 (*p* = 0.000) **	r = −0.445 (*p* = 0.001) **	r = −0.409 (*p* = 0.002) **	r = −0.494 (*p* = 0.000) **
RAa (mm^2^)	2761.05 ± 912.31	r = 0.067 (*p* = 0.622)	r = −0.003 (*p* = 0.982)	r = −0.043 (*p* = 0.754)	r = 0.192 (*p* = 0.185)
LVa (mm^2^)	2425.86 ± 779.36	r = −0.155 (*p* = 0.254)	r = 0.267 (*p* = 0.049) *	r = −0.120 (*p* = 0.381)	r = −0.124 (*p* = 0.398)
RVa (mm^2^)	3210.62 ± 830.67	r = 0.257 (*p* = 0.055)	r = 0.262 (*p* = 0.054)	r = 0.191 (*p* = 0.163)	r = 0.182 (*p* = 0.210)

Note. AAd, diameter of ascending aorta; LAa, left atrium area; LAd, diameter of left atrium; LPAd, diameter of left pulmonary artery; LVa, left ventricular area; LVd, diameter of left ventricle; MPAd, diameter of main pulmonary artery; MPAd/AAd, the ratio of diameter of main pulmonary artery to diameter of ascending aorta; RAa, right atrium area; RAd, diameter of right atrium; RAd/LAd, the ratio of diameter of right atrium to diameter of left atrium; RPAd, diameter of right pulmonary artery; RVa, right ventricular area; RVd, diameter of right ventricle; RVd/LVd, the ratio of diameter of right ventricle to diameter of left ventricle. * *p* < 0.05; ** *p* < 0.001.

**Table 5 jcm-12-01297-t005:** ICC of different models and pressure types.

	95% Confidence Interval			F Test with True Value 0	
Task and Method	ICC ^b^	Lower Bound	Upper Bound	Value	Sig.
mPAP and CatBoost					
Single Measures	0.525 ^a^	−0.289	0.898	3.211	0.091
Average Measures	0.689 ^c^	−0.813	0.946	3.211	0.091
mPAP and SVM					
Single Measures	0.074 ^a^	−0.668	0.742	1.160	0.431
Average Measures	0.138 ^c^	−4.015	0.852	1.160	0.431
mPAP and XGBoost					
Single Measures	0.877 ^a^	0.447	0.978	15.211	0.002
Average Measures	0.934 ^c^	0.617	0.989	15.211	0.002
dPAP and XGBoost					
Single Measures	0.824 ^a^	0.280	0.967	10.355	0.006
Average Measures	0.903 ^c^	0.438	0.983	10.355	0.006
sPAP and XGBoost					
Single Measures	0.963 ^a^	0.803	0.994	53.139	0.000
Average Measures	0.981 ^c^	0.890	0.997	53.139	0.000
TPR and XGBoost					
Single Measures	0.521 ^a^	−0.294	0.897	3.173	0.093
Average Measures	0.685 ^c^	−0.834	0.946	3.173	0.093

Note. Two-way mixed effects model where people effects are random and measures effects are fixed; ^a^ The estimator is the same, whether the interaction effect is present or not; ^b^ Type C intraclass correlation coefficient using a consistency definition. The between-measure variance is excluded from the denominator variance; ^c^ This estimate is computed assuming the interaction effect is absent, because it is not estimable otherwise.

## Data Availability

Data is unavailable due to privacy.

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
