# Peer review of "Machine Learning Based on Computed Tomography Pulmonary Angiography in Evaluating Pulmonary Artery Pressure in Patients with Pulmonary Hypertension"

_jcm, 2023, doi:10.3390/jcm12041297_

Round 1

Reviewer 1 Report

Comments for authors

This reviewer has read with the article written by Nan Zhang, et al. entitled “Machine Learning Based on Computed Tomography Pulmonary Angiography in Evaluating Pulmonary Artery Pressure in Patients with Pulmonary Hypertension”. In this article, the authors evaluate evaluate the value of CTPA morphological data analysis in the diagnosis of PH. As a result, the proposed machine learning framework on CTPA enables accurate segmentation of pulmonary artery and heart and automatic assessment of the PAP parameters and has capability to accurately distinguish different PH patients with mPAP and sPAP.

I have annotated the manuscript with several minor corrections, which I believe will improve the readability of the paper.

Major Comment

1.     In this study, the analysis was performed only for PH patients, but how about adding patients who underwent right heart catheterization and CTPA without PH? The reviewer suggests that adding this analysis might lead to more universal research results.

Minor Comment

1. It is not appropriate to list all variables in the same way in a table. Please revise the notation method as mean + SD for continuous variables with normal distribution, median [range] for continuous variables with non-normal distribution, and number(percent) for categorical variables.

2. In all tables, please list the explanations of the abbreviations in alphabetical order.

3. The analysis performed in this study should not be described only in the Result section but should also be explained in the Method section. (e.g., Dice score, Bland-Altman analysis)

4. In the Results section, the authors mention the difference between Figures 4(b) and 4(c), but the difference between the two images is difficult to understand. Please modify the contents of the figure so that it is easier for the reader to understand.

Author Response

I have annotated the manuscript with several minor corrections, which I believe will improve the readability of the paper.

Response: Thank you for your positive comments and thorough review. Your comments and suggestions are constructive and very helpful for us to improve the quality of our manuscript.

Major Comment

Reviewer Comment C1.1: In this study, the analysis was performed only for PH patients, but how about adding patients who underwent right heart catheterization and CTPA without PH? The reviewer suggests that adding this analysis might lead to more universal research results.

Response R1.1: Thank you for this constructive suggestion. It is indeed a clinical problem to identify the PH patients with CTPA images, especially for differentiation of the pulmonary valve stenosis patients, PH patients with mild pulmonary pressure elevated, newly PH patients with critical morphological changes and normal people. We proposed to enroll health people in this study during experimental design stage. Unfortunately, in our single center, the sample size of normal people underwent right heart catheterization and CTPA simultaneously was not enough for analysis. So, finally, we only enrolled PH patients in the study. This was also discussed as a disadvantage in “Discussion”. In PH patients, PAP parameters have important clinical significance for diagnosis, severity evaluation and prognosis prediction. The proposed machine learning framework on CTPA enables accurate segmentation of pulmonary artery and heart and automatic assessment of the PAP parameters in PH patients. The capability of machine learning in differentiation of normal people and PH patients will be evaluated in future study.

 Minor Comment

Reviewer Comment C1.1: It is not appropriate to list all variables in the same way in a table. Please revise the notation method as mean + SD for continuous variables with normal distribution, median [range] for continuous variables with non-normal distribution, and number(percent) for categorical variables.

Response R1.1: Following your suggestion, we have now modified the notation method of different data. Please refer to the revised manuscript.

Reviewer Comment C1.2: In all tables, please list the explanations of the abbreviations in alphabetical order.

Response R1.2: Following your suggestion, we have now list the explanations of the abbreviations in alphabetical order beneath each table. Please refer to the revised manuscript.

Reviewer Comment C1.3: The analysis performed in this study should not be described only in the Result section but should also be explained in the Method section. (e.g., Dice score, Bland-Altman analysis)

Response R1.3: Following your suggestion, we have now add “We used dice score to evaluate the performance of the proposed segmentation framework compared with the nnU-Net.” in “2.8. Statistics Analysis” section. Please refer to the revised manuscript.

Reviewer Comment C1.4: In the Results section, the authors mention the difference between Figures 4(b) and 4(c), but the difference between the two images is difficult to understand. Please modify the contents of the figure so that it is easier for the reader to understand.

Response R1.4: In figure 4(b), the segmentation mask of LV occupied LV area and myocardium of whole LV free wall and part of interventricular septum. And the segmentation mask of LV was correct in figure 4(C). We have now added “Compared with the result of proposed network framework in figure 4(c), myocardium of LV is mis-segmented as chamber of LV using nnU-Net in figure 4(b) which overestimates the area of LV.” in the “Result”.

Reviewer 2 Report

Comments to the Author

The manuscript entitled: “Machine learning based on computed tomography pulmonary angiography in evaluating pulmonary artery pressure in patients with pulmonary hypertension” described the development of a framework to evaluate pulmonary artery pressure through machine learning based on CTPA. Although the authors showed interesting findings, I would like to raise the following concerns.

Q1.

Right heart catheterization is considered the gold standard for the evaluation of pulmonary hypertension, as it provides direct measurement of pulmonary artery pressure. However, echocardiography is widely regarded as the most prevalent non-invasive method for evaluating pulmonary artery pressure in clinical practice. Patients undergoing CTPA examinations are at risk for certain adverse effects, such as exposure to radiation and the administration of contrast agents. Please comment on the advantages of utilization of CTPA in contrast to echocardiography.

Q2.

There are variations in chamber size depending on the underlying cardiac pathology. For example, in dilated cardiomyopathy, the chamber size is dilated, but the pulmonary artery pressure remains within the normal range unless there is congestion. Conversely, constrictive pericarditis can be associated with considerable pulmonary hypertension even when the lumen size is unremarkable. What is the distribution of underlying cardiac pathology in this study?

Q3.

Is it feasible to assess cardiac chamber size in terms of volume rather than area? For instance, when evaluating left atrial loading, left atrial volume is deemed more precise than left atrial area.

Author Response

Reviewer Comment C2.1: Right heart catheterization is considered the gold standard for the evaluation of pulmonary hypertension, as it provides direct measurement of pulmonary artery pressure. However, echocardiography is widely regarded as the most prevalent non-invasive method for evaluating pulmonary artery pressure in clinical practice. Patients undergoing CTPA examinations are at risk for certain adverse effects, such as exposure to radiation and the administration of contrast agents. Please comment on the advantages of utilization of CTPA in contrast to echocardiography.

Response R1.1: Echocardiography is widely used in PH patient for non-invasive quantitative evaluation of pulmonary artery pressure as well as cardiac morphological and functional parameters. Because of large field of view in CTPA examination, the intra-pulmonary and mediastinal structures can be displayed. And by using the isotropic volumetric data with high spatial resolution, the cardiac and vascular structures can be evaluated in multiple orientation and reconstruction modes. With contrast media enhancement, the vascular structure can be separated clearly with surrounding soft tissue. So by using CTPA data, we can assess different pathogenesis of PH patients, the severity of relevant pulmonary vascular and lung lesions, and the pulmonary vascular remodeling in PH patients, which cannot be completed with echocardiography only.  Reviewer Comment C2.2: There are variations in chamber size depending on the underlying cardiac pathology. For example, in dilated cardiomyopathy, the chamber size is dilated, but the pulmonary artery pressure remains within the normal range unless there is congestion. Conversely, constrictive pericarditis can be associated with considerable pulmonary hypertension even when the lumen size is unremarkable. What is the distribution of underlying cardiac pathology in this study?

Response R2.2: Sorry for not providing a clear description in this study. None of the 55 PH patients combined with cardiomyopathy as well as constrictive pericarditis. The most common reason for PH was chronic pulmonary embolism (15/55, 27.27%), with idiopathic PH (10/55, 18.18%), connective tissue disease-associated PH (8/55, 14.55%), pulmonary heart disease (6/55, 10.91%), left to right shunt congenital heart disease (4/55, 7.27%) as followed. And besides the morphological changes of bilateral atrium and ventricle, main pulmonary artery and bilateral pulmonary artery trunk are considered simultaneously in the framework to improve the accuracy of PAP parameter evaluation.  

Reviewer Comment C2.3: Is it feasible to assess cardiac chamber size in terms of volume rather than area? For instance, when evaluating left atrial loading, left atrial volume is deemed more precise than left atrial area.

Response R2.3: Thank you for this constructive suggestion. The volumetric information could improve the accuracy of PAP parameter evaluation with CTPA data with this proposed machine learning framework. CTPA images allow precise 3D analysis of four cardiac chambers and whole pulmonary arteries above segmental level with isotropic character, high spatial and density resolution. Our study only included one-dimensional and two-dimensional morphological features of heart and pulmonary arteries and the ratio between selected features, such as RAd/LAd, LVd and LAa. This was the disadvantage of this study which was mentioned in the “Discussion”. In the future study, we will explore the volumetric information of heart and pulmonary arteries morphology as well as spatial relationship between different intra- and extra-cardiac structures to improve the accuracy of PAP parameter evaluation. See lines 447-449 and 488-493.